# Machine Learning for Risk Prediction of Oesophago-Gastric Cancer in Primary Care: Comparison with Existing Risk-Assessment Tools

**DOI:** 10.3390/cancers14205023

**Published:** 2022-10-14

**Authors:** Emma Briggs, Marc de Kamps, Willie Hamilton, Owen Johnson, Ciarán D. McInerney, Richard D. Neal

**Affiliations:** 1School of Computing, University of Leeds, Leeds LS2 9JT, UK; 2Leeds Institute for Data Analytics, University of Leeds, Leeds LS2 9NL, UK; 3The Alan Turing Institute, London NW1 2DB, UK; 4Department of Health and Community Sciences, University of Exeter, Exeter EX1 2LU, UK; 5Academic Unit of Primary Medical Care, University of Sheffield, Sheffield S10 2TN, UK

**Keywords:** early detection, cancer diagnosis, electronic health record, machine learning, oesophago-gastric cancer, primary care, risk-assessment

## Abstract

**Simple Summary:**

Oesophago-gastric cancer is one of the commonest cancers worldwide, yet it can be particularly difficult to diagnose given that initial symptoms are often non-specific and routine screening is not available. Cancer risk-assessment tools, which calculate cancer risk based on symptoms and other risk factors present in the primary care record, can aid decisions on referrals for cancer investigations, facilitating earlier diagnosis. Diagnosing common cancers earlier could help improve survival rates. Using UK primary care electronic health record data, we compared five different machine learning techniques for probabilistic classification of cancer patients against a current widely used UK primary care cancer risk-assessment tool. The machine learning algorithms outperformed the current risk-assessment tool, with a higher overall accuracy and an ability to reasonably identify 11–25% more cancer patients. We conclude that machine-learning-based risk-assessment tools could help better identify suitable patients for further investigation and support earlier diagnosis.

**Abstract:**

Oesophago-gastric cancer is difficult to diagnose in the early stages given its typical non-specific initial manifestation. We hypothesise that machine learning can improve upon the diagnostic performance of current primary care risk-assessment tools by using advanced analytical techniques to exploit the wealth of evidence available in the electronic health record. We used a primary care electronic health record dataset derived from the UK General Practice Research Database (7471 cases; 32,877 controls) and developed five probabilistic machine learning classifiers: Support Vector Machine, Random Forest, Logistic Regression, Naïve Bayes, and Extreme Gradient Boosted Decision Trees. Features included basic demographics, symptoms, and lab test results. The Logistic Regression, Support Vector Machine, and Extreme Gradient Boosted Decision Tree models achieved the highest performance in terms of accuracy and AUROC (0.89 accuracy, 0.87 AUROC), outperforming a current UK oesophago-gastric cancer risk-assessment tool (ogRAT). Machine learning also identified more cancer patients than the ogRAT: 11.0% more with little to no effect on false positives, or up to 25.0% more with a slight increase in false positives (for Logistic Regression, results threshold-dependent). Feature contribution estimates and individual prediction explanations indicated clinical relevance. We conclude that machine learning could improve primary care cancer risk-assessment tools, potentially helping clinicians to identify additional cancer cases earlier. This could, in turn, improve survival outcomes.

## 1. Introduction

Oesophago-gastric cancer is one of the commonest cancers worldwide [1], with approximately 15,800 cases diagnosed annually in the UK (statistics from 2019) [2,3]. It is also associated with some of the poorest survival rates, in part attributable to late-stage diagnosis [4]. International comparisons highlight particularly inferior cancer survival rates in the UK amongst economically developed countries [5]. When comparing cancer survival rates across stage at diagnosis, five-year net survival for oesophago-gastric cancer in the UK is up to 65.0% when diagnosed at the earliest stage, compared with between 16.0–25.0% at stage 3, and as low as 2.0% at stage 4 [6].

Diagnosing oesophago-gastric cancer can be difficult and mainly depends upon presentation with symptoms to primary care, given that there is no routine screening besides the monitoring of patients with Barrett’s oesophagus [1]. Often, patients present with non-specific symptoms indiscernible from symptoms of more minor issues, with the two main alarm symptoms—dysphagia and iron-deficiency anaemia—only accounting for a small amount of initial presentations [7]. Diagnostic intervals are generally much greater for patients with no alarm symptoms, usually those with early-stage and localised disease [8].

The latest long-term plan for the National Health Service (NHS) in England, launched in 2019, aspires to diagnose 75.0% of cancers in England at stage 2 or earlier by 2028. Taking steps towards improving diagnostic performance for oesophago-gastric cancer would help meet this objective [9]. Furthermore, expediting diagnosis of symptomatic patients may have benefits for both mortality and morbidity [10,11].

Clinical decision-support tools can assist primary care clinicians in diagnosing cancer earlier [12]. Being able to more accurately detect warning signs and symptoms of oesophago-gastric cancer to strategically refer high-risk patients for endoscopic tests could see great benefits [7]. Additionally, adopting a more ‘personalised’ approach, as opposed to a population-based approach, to cancer risk assessment and staging could be a valuable approach in helping to raise survival rates [13].

Current UK primary care interventions to accelerate cancer diagnosis include cancer Risk-Assessment Tools (RATs): a series of 2 × 2 matrices which display risk scores associated with individual and pairwise combinations of diagnostic factors based on positive predictive values [14,15]. These were derived from the CAncer Prevention in ExetER studies (CAPER) [16] published in 2009: a series of case-control studies designed to quantify the risk of certain cancers using diagnostic risk factors available in the primary care record, such as symptoms and lab test results. Another predominant diagnostic tool is QCancer^®^ [17], first created in 2012. The QCancer^®^ tool consists of two risk-prediction algorithms (one for men and one for women) which generate individual risk scores based on symptomatic, baseline, and lifestyle-related diagnostic factors across 13 different types of common cancer [18,19]. These studies have been instrumental in influencing current UK guidelines on recognising and referring suspected cancer cases in primary care [20]. Efforts have been made to digitise these tools; however, the tools themselves remain in relative infancy and have certain limitations: the risk scores generated by the oesophago-gastric RAT (ogRAT), for example, take into account a maximum of two symptoms simultaneously, and only give actionable risk scores for patients over 55 [7]. Additionally, sets of diagnostic factors used in QCancer^®^ are inexhaustive [18,19] and the oesophago-gastric QCancer^®^ model may have a tendency to over-predict high risk scores [21]. At the time of writing, the Electronic RIsk assessment for CAncer (ERICA) trial is underway to assess the clinical usefulness of digitised RATs integrated into major primary care clinical systems [22].

Given the heterogeneous nature of cancer, using simple scoring techniques, relying on known diagnostic factors, or placing more emphasis on alarm symptoms are likely to be insufficient strategies to assess risk comprehensively [8,23]. Machine Learning (ML) may be able to elevate diagnostic performance, exploiting large-scale electronic health record (EHR) data with diverse feature sets to automatically determine more individually tailored risk scores [24]. However, attempts so far lack clinical validation, and major challenges—such as improving robustness, interpretability, and clinical relevance—abound [24]. The vast majority of efforts employing ML for cancer diagnosis have concentrated on improving diagnostic accuracy for medical imaging [25,26], with few studies performed on primary care data [26]. This is despite evidence suggesting that some of the greatest benefits could be seen by reducing diagnostic delays at the point of primary care [27].

To the extent of the authors’ knowledge, based on our investigation of the literature, to date there has been no direct comparison of ML-based risk-assessment tools for cancer diagnosis in primary care against currently implemented tools in the UK [26,28]. Furthermore, there has been a distinct lack of focus on upper gastrointestinal cancers—including oesophago-gastric cancer—in ML-based clinical risk-prediction tools [28]. The aim of this study, therefore, was to assess suitability of an ML-based approach, by benchmarking the performance of probabilistic, supervised, ML classifiers against the UK Cancer ogRAT for risk prediction of oesophago-gastric cancer, using data derived from primary care EHRs.

The study seeks to address the question as to whether ML models could help with the strategic referral of patients for further investigation and the earlier detection of a greater number of cancer cases. In terms of model performance, this translates to the extent to which recall values might be increased whilst roughly preserving precision values, thereby detecting a larger number of cases yet minimising the number of unnecessary procedures.

We focus on traditional ML approaches, which can perform comparatively well in the context of machine learning and deep learning [24,29] and are therefore favoured for the purpose of this study in the interest of minimising computational complexity and supporting interpretability [29]. Explainability and interpretability should be emphasised from the outset to allow greater insight into clinical relevance of models and garner clinician trust and understanding [30,31]. The UK Department of Health and Social Care recommends that such qualities are built into data-driven health technologies by design in order to maximise transparency [31].

## 2. Materials and Methods

### 2.1. Data Overview

The dataset employed in this study was a primary care EHR dataset, derived from a 2013 case-control study investigating risk factors of oesophago-gastric cancer, the results of which were used to develop the ogRAT model [7]. The data originated from the General Practice Research Database (GPRD) in the UK, a high-quality database containing anonymised EHR data on approximately 4.8 million patients from around 600 general practices across the UK [32,33].

The resulting dataset comprises information on 40,348 patients (7471 cases and 32,877 controls), selected from the database using a list of 42 tumour diagnostic codes, with up to five (mean of four) matched controls (matched according to age, sex, and practice) for each case [7]. Inclusion criteria consisted of a diagnosis during or after the year 2000, age of ≥40 years, and availability of consultations in the year prior to diagnosis of the case [7]. All medical evidence was collected in the year prior to diagnosis. Variables include basic demographic information, ICPC-coded symptoms, and various lab test results (See Appendix A for further information). Repeat presentations of symptoms (up to one repeat, where available) were also recorded. The mean age of patients was 73 years (±standard deviation of 10), with a 64.2/35.8% split of male and female patients respectively. Oesophageal cancer diagnoses constituted 65.3% of the cases, with 34.7% being gastric cancer cases (which roughly corresponds to the national incidence rate [34]). See Table 1 for the full list of patient characteristics.

Regarding pre-processing, much of this had already been performed prior to acquisition of the dataset, including the generalisation of potentially identifiable information such as age and date of birth [7]. Symptoms had been one-hot encoded, and all lab test results with continuous values were converted to binary (normal/abnormal), except for cholesterol (which was included as both a categorical and numerical variable) [7]. Additional pre-processing performed specifically for this study involved min-max normalisation of continuous variables and conversion of binary categorical (string) variables into dummy variables (0 or 1).

### 2.2. Ethical Approval

This secondary data analysis study was performed in line with the original ethical approval remit. This was approved by the Independent Scientific Advisory Committee (ISAC). The dataset does not contain any personally identifiable information.

### 2.3. Feature Selection

Initial feature selection was based on known diagnostic factors for oesophago-gastric cancer [35,36,37] and basic demographic variables (age, gender), to be included as baseline. Additional features were selected following a series of experiments with different subsets: initially constraining the set to just those present in the ogRAT, and then testing combinations including other symptoms. The final selection of features corresponded to the best-performing model on the merit of model performance and clinical suitability of predictions. Model-based feature selection fine-tuned this set further.

### 2.4. Predictive Models

Five probabilistic ML classifiers were chosen: Random Forest, Support Vector Machine (with probability calibration using Platt scaling), Logistic Regression, Naïve Bayes, and Extreme Gradient Boosted Decision Trees. These are commonly used ML techniques which have been shown to be successful for other clinical risk-prediction tasks [24,29].

Model hyperparameters were fine-tuned using fivefold grid search cross-validation (see Appendix A). The problem was formulated as a binary classification task between ‘cancer’ (case) and ‘no cancer’ (control), with a corresponding probability of cancer given.

### 2.5. Evaluation

The dataset was split into a train/test split of 75.0%/25.0%, respectively, allocated randomly. Fivefold cross-validation on the training set was used for model selection with the final performance evaluated on the unseen test set.

Classification performance was evaluated on the basis of commonly used ML performance metrics including overall accuracy, precision, recall, F1 score, and Area Under the Receiver Operating Characteristic curve (AUROC). Particular attention was paid to the trade-off between precision and recall as the aim was to build models that optimise the proportion of cancer cases detected (maximising recall) whilst simultaneously minimising the proportion of unnecessary referrals (maximising precision). Using these performance measures, the ML models’ predictions were compared against the ogRAT’s predictions on the same test set. Clinical justifiability was assessed using prediction explanations and feature importance estimation.

Data was processed using the popular software development toolkit Jupyter Notebooks in the programming language Python (v3.7.10) using the pandas (v1.2.4), scikit-learn (v0.24.2), xgboost (v1.5.0), and numpy (v1.20.2) libraries for model building, training, testing, and performance evaluation; matplotlib (v3.3.4) to generate plots; and the Local Interpretable Model-Agnostic Explanations (LIME) package for model explanations [38].

## 3. Results

### 3.1. Model Performance

All five ML approaches classified oesophago-gastric cancer cases with a high level of accuracy, surpassing that of the ogRAT. All models achieved an accuracy of 0.89: the accuracy of the ogRAT when employed at the typical 2.0% threshold came in slightly lower at 0.87. The highest-performing models in terms of AUROC were Linear Support Vector Machine, Logistic Regression, and XGBoost (0.87 AUROC). Table 2 shows the performance of all models in comparison with the ogRAT when employed at the 1.0% or 2.0% risk threshold for cancer (the typical thresholds at which the ogRAT was designed to be used). See Appendix A for results of the full performance evaluation for all models across a range of classification thresholds. The classification threshold is flexible; a couple of examples are given here for each model (Table 2) to demonstrate thresholds at which performances are comparable to the ogRAT—in terms of precision and recall—and overall accuracy is maximised. The aim was to improve upon overall accuracy and surpass the current recall rate offered by the ogRAT model—which equates to detecting a larger proportion of cases—whilst forfeiting as little as possible in terms of precision, thereby limiting false positives. The recall rate offered by the current risk-assessment strategy, the ogRAT, stands at 33.0% when adopting the typical 2.0% risk threshold (Table 2).

For example, employing Logistic Regression at the 0.425 threshold improves upon the accuracy and AUROC score of the ogRAT, with a dramatic increase in recall of cancer patients (potentially recalling, in absolute terms, from 17.0% up to 25.0% more cancer patients depending on the chosen risk threshold for the ogRAT) and little cost to precision. Even comparing the performance with the most lenient (1.0%) risk boundary for the ogRAT results in the successful detection of 17.0% more cancer patients with little cost to precision (a 5.0% reduction) and a slight improvement in overall accuracy (Table 2). Adjusting the Logistic Regression risk threshold to 0.75 to roughly preserve the higher false positive rate of the ogRAT would still see the identification of up to 11.0% more cases (Table 2).

Table 3 displays the recall offered by two of the best-performing ML models (Linear Support Vector Machine and Logistic Regression) when exactly preserving the precision rate of the ogRAT model. Using these precision values as fixed reference points, it is clear that the ML models could offer a greater recall (between 8.0–11.0% higher) at the same precision as the ogRAT. This translates to the potential to detect a greater number of cancer cases without the need for any extra investigations in the cancer-free population.

### 3.2. Feature Contribution Estimates

The final feature set included all 16 previously acknowledged risk factors from the Stapley et al. (2013) study [7], with the addition of age (categorised as ≤55 and >55), and serum cholesterol level. Figure 1 shows the estimated importance rankings corresponding to the Support Vector Machine (linear) model, for this feature set. From the feature importance estimates (Appendix A), it appears that all models assigned a similar ranking to the features, with dysphagia (considered to be the classical ‘alarm’ symptom for OG cancer) accruing the highest risk contribution across all models, closely followed by epigastric pain, weight loss, dyspepsia, and reflux, in varying order. The remaining features were assigned similar importance across all models, with one notable exception being cholesterol level: for the Random Forest model, high cholesterol corresponded to increased cancer risk; however, Support Vector Machine and Logistic Regression associated low cholesterol levels with elevated OG cancer risk. Ostensibly, this seems inaccurate, but appears to be consistent with the medical literature which suggests that abnormally low levels of low-density lipoprotein cholesterol—and in turn, serum cholesterol—are in fact associated with increased OG cancer risk [39,40].

## 4. Discussion

We developed and compared five ML-based probabilistic classifiers for risk prediction of oesophago-gastric cancer using data derived from the primary care EHR consisting of basic demographic information, symptomatic evidence, and various lab test results. The performance of these models was benchmarked against an oesophago-gastric cancer risk-assessment tool—the ‘ogRAT’—currently employed in primary care clinics across the UK.

Our models selected a combination of previously established clinical features in oesophago-gastric cancer risk-prediction models [7,20] and other less commonly used diagnostic factors [39,40] to deliver predictions with a high overall accuracy (0.89 for Linear Support Vector Machine). The best-performing techniques were Linear Support Vector Machine, Logistic Regression, and XGBoost, but all techniques displayed similar performances. Models performed relatively consistently across demographic groups (according to age and sex) and were slightly better performing for oesophageal cases than gastric cases, as was also observed in the ogRAT model (see Appendix A for results stratified according to demographic groups and cancer sites).

Feature importance also appeared similar across techniques, with dysphagia consistently established as the biggest contributor to risk (see Figure 1 for the feature importance ranking estimate for Linear Support Vector Machine, and Appendix A for all other models). Overall, models showed good calibration to the test dataset (Appendix A), with the Naïve Bayes model being notably the least well calibrated. It will be more enlightening to assess aspects such as calibration and clinical utility during further validation phases on external datasets [42], in order to verify that these characteristics are maintained in general.

Such models could be highly influential in meeting current challenges faced in risk prediction for common cancers in primary care. One major challenge is being able to strategically select a greater number of patients for referral for cancer investigation, achieving an optimal balance between detection and conversion rates [7]. ML-based models could help achieve this: the current recall rate at the point of primary care is low, with only 31.0% of oesophago-gastric cancer cases identified by stage 2 and 34.0% of all cases missed completely by primary care clinicians [43]. The current recommended risk threshold for urgent investigation of suspected oesophago-gastric cancer is 3.0% [20]; however, there have been suggestions to potentially liberalise this threshold to 2.0%, or even 1.0%, in order to detect more cases earlier [7,44]. For oesophago-gastric cancer, increasing the referral rate to allow for the detection of a greater number of cases needs to be done in such a way which minimises any resulting increase in false positive rates [44]. From the results of this study, using the ogRAT model at the recommended risk threshold of 3.0% would detect around 31.0% of cases from symptomatic presentation to primary care (Appendix A). Employing the ogRAT at the more liberal threshold of 2.0% would encourage the referral of around 33.0% of cases. ML could increase this further: detecting, in this study, around 58.0% of these cases (with a slight cost to precision), 44.0% if roughly preserving the false positive rate (a decrease of 1.0% in precision), or 41.0% if exactly preserving the false positive rate. Achieving an overall increase in recall of 10.0%, for example, would theoretically equate to the earlier detection of over 1500 additional oesophago-gastric cancer cases in the UK annually [2,3].

Investigation into individual prediction explanations using LIME [38] highlighted cases which ML models would flag as high risk of cancer whereas the ogRAT would not, revealing that ML picks up on certain cases with vaguer symptoms which would otherwise go undetected using RATs (see Appendix A for one such example). Results also showed that ML could improve upon areas in which RATs are prone to over-generalisation: for example, the ogRAT would not flag any cases in under-55s, who are deemed collectively to have a low risk [7].

Therefore, integrating these ML-based classifiers into primary care cancer risk-prediction tools could assist in prioritisation of suspected cancer patients for referral. Prompting early testing could expedite diagnosis and treatment, and could improve outcomes in terms of both mortality and morbidity [45]. Such a tool would, for example, take the form of a digitised risk-prediction tool embedded in the primary care clinician’s workflow. Based on symptomatic and lab test information input with clinical codes such as SNOMED or Read codes, the ML algorithm would calculate a cancer risk score for each patient and subsequently flag patients with a high risk score according to a predetermined risk threshold. Implementing the tools in this way does risk bias related to coding inconsistencies and unreported symptoms, for example, and future work should seek to address the mitigation of such biases. Determining an optimal risk threshold is not a trivial task and requires a clinical utility assessment to quantify the trade-off between detecting more cases and minimising false positives [42,46]. The extent to which an increase in recall is worth a slight cost to precision is currently unclear. Risk thresholds are easily adjustable depending on the clinical context and should be initiated with a view to maximising clinical benefits (for example, in terms of costs and test harms) [42,46].

Regarding current progress with implementation, cancer risk-assessment tools are not currently widely used in primary care and there are significant hurdles to increasing their adoption [47]. As of 2019, cancer risk tools were available to approximately a third of all primary care practices in the UK, with 18.5% of practices having access to these in electronic form [47]. However, they are currently underused: only 16.7% of practices were likely to use any form of cancer risk-assessment tool [47]. Likely explanations for this profound underuse include an aversion to using them due to time constraints during consultations and ‘prompt fatigue’, whereby clinicians become overburdened with on-screen alerts [48]. Another reason is an overall lack of awareness about these tools [48,49]. This is despite the fact that many clinicians find them very helpful, and the tools have the potential to assist them in cases with vaguer presentations, influence their decisions on referral, and serve as a reminder of the possibility of cancer [49]. Furthermore, they have been shown to have an educational effect [12,48], with one study finding that the use of cancer risk-assessment tools gradually improved primary care clinicians’ decision-making over time [12]. This suggests that incorporating new electronic cancer decision-support tools into primary care clinical systems could prove very valuable provided that they are designed and implemented in such a way which supports clinicians’ work processes, and that awareness and education surrounding such tools is prioritised. Finding optimal ways of effectively integrating such tools remains challenging [48].

### Limitations

Availability of data posed certain limitations. Firstly, although the dataset includes all the key known diagnostic factors for oesophago-gastric cancer, it does not provide a comprehensive set of all potential diagnostic factors, omitting symptomatic factors such as collapse or aetiological factors such as gastritis or obesity, as highlighted in other pieces of medical literature [23,50]. Furthermore, other types of data are not included, such as lifestyle data, family history, or previous surgical procedures, all of which could contribute to oesophago-gastric cancer risk [23,51]. Additionally, the coarse granularity of the majority of the feature-variable values restricts the precision of predictions. Limited availability of demographic variables is also an issue. In order to avoid discriminatory biases moving forward, variables such as socioeconomic deprivation and racial characteristics should also be taken into consideration [52,53]. Including other characteristics of symptomatic presentations, such as duration or severity of symptoms, could also improve predictive performance [54].

Limitations regarding performance evaluation were also experienced. Firstly, comorbidities and other cancers were not accounted for and the models’ only task was to differentiate between the presence and absence of oesophago-gastric cancer. Quantifying how much earlier these tools might be able to detect cases using temporal and/or staging information will be an essential next step. Assessing performance in comparison with other frontrunners for oesophago-gastric risk assessment, such as QCancer^®^ [55], will also be important. Naturally, any clinical benefits of such models would be highly dependent on model–clinician interaction, which forthcoming implementation trials such as the ERICA trial [22] may shed some light on. It is important to note that implementation phases should emphasise patient safety, adopting a systems approach to risk, safety, and accident avoidance, particularly since current regulations of health informatics solutions are likely insufficient [56].

## 5. Conclusions

In conclusion, these results favour the use of an ML-based approach for primary care risk prediction for oesophago-gastric cancer. These risk-prediction models could assist GPs in diagnostic decision-making and highlight potential cancer cases which might otherwise be missed. Such techniques could potentially be extended to other common—and possibly uncommon—cancers. These would be particularly useful for those cancers typically expressed by non-specific symptoms and with a lack of available routine screening. External validation, comparison against other existing cancer risk-prediction tools, and incorporation of a more diverse feature set involving other indicators of health are necessary moving forward.

Assessing the performance of models in combination with clinicians will be imperative to gauge practicality and usefulness in a real-world scenario. Future iterations of AI-enhanced cancer risk-prediction models could involve dynamic versions which update in real time as new information is accumulated, or sequential models exploiting temporal data and accounting for time intervals between symptoms. Implementation of usable decision-support tools could be extended to include algorithms which better reflect uncertainty, dynamically adapt to new information, and incorporate additional functionality such as user prompts for additional information to improve the quality of predictions.

## Figures and Tables

**Figure 1 cancers-14-05023-f001:**
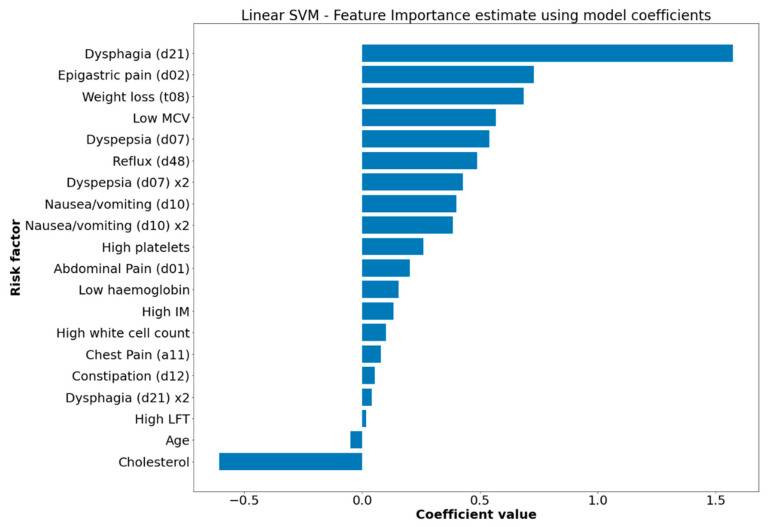
Feature importance graph displaying relative contribution of diagnostic factors for oesophago-gastric cancer to model risk score. Model: Support Vector Machine (with linear kernel). Feature weights represented by model coefficients. Generated with matplotlib [41].

**Table 1 cancers-14-05023-t001:** Patient characteristics of 40,348 individuals according to basic demographic information, symptoms, lab test results, and cancer site, organised by case/control. Figures given as number of occurrences (in total cohort/case group/control group) and percentages.

Patient Characteristic	Total Cohort Count (%) N = 40,348	Case Count (%) N = 7471 (18.5%)	Control Count (%) N = 32,877 (81.5%)
**Age**			
Under 55	2550 (6.3)	514 (6.9)	2036 (6.2)
Over 55	37,798 (93.7)	6957 (93.1)	30,841 (93.8)
**Sex**			
Male	14,860 (36.4)	2672 (35.8)	12,188 (37.1)
Female	25,488 (63.6)	4799 (64.2)	20,689 (62.9)
**Cancer Site**			
Oesophagus	26,360 (65.3)	4854 (65.0)	21,506 (65.4)
Stomach	13,988 (34.7)	2617 (35.0)	11,371 (34.6)
**Symptoms**			
Abdominal Pain	2215 (5.5)	905 (12.1)	1310 (4.0)
Chest Pain	2316 (5.7)	727 (9.7)	1589 (4.8)
Constipation	1681 (4.2)	608 (8.1)	1073 (3.3)
Cough	4782 (11.9)	1005 (13.4)	3777 (11.4)
Dyspepsia	2085 (5.2)	1294 (17.3)	764 (2.3)
Dyspepsia (repeat)	699 (1.7)	532 (7.1)	167 (0.5)
Dysphagia	2605 (6.5)	2420 (32.3)	185 (0.6)
Dysphagia (repeat)	678 (1.7)	635 (8.5)	43 (0.1)
Epigastric Pain	883 (2.2)	617 (8.3)	266 (0.8)
Fatigue	1362 (3.4)	388 (5.2)	974 (3.0)
Nausea/Vomiting	1616 (4.0)	979 (13.1)	637 (1.9)
Nausea/Vomiting (repeat)	534 (1.3)	386 (5.2)	148 (0.5)
Reflux	1355 (3.4)	842 (11.3)	513 (1.6)
Shortness of breath	2621 (6.5)	629 (8.4)	1992 (6.1)
Weight loss	891 (2.2)	615 (8.2)	276 (0.8)
**Lab Test Results**			
Cholesterol (high)	6100 (15.1)	920 (12.3)	5180 (15.8)
Haemoglobin (low)	5398 (13.3)	2045 (27.3)	3353 (10.2)
Inflammatory Markers (high)	2431 (6.0)	1010 (13.5)	1421 (4.3)
Liver Function Test (high)	4751 (11.8)	1272 (17.0)	3479 (10.6)
Mean Corpuscular Volume (low)	1007 (2.5)	640 (8.6)	367 (1.1)
Platelet Count (high)	1274 (3.2)	706 (9.4)	568 (1.7)
White Cell Count (high)	1533 (3.8)	671 (9.0)	862 (2.6)

**Table 2 cancers-14-05023-t002:** Performances of machine-learning-based probabilistic classifiers on test dataset in comparison with oesophago-gastric cancer risk-assessment tool (ogRAT) matrix (performance calculated on same test set) for prediction of oesophago-gastric cancer incidence.

Classifier	AUROC	Classification Threshold	Accuracy	Precision	Recall	F1
Linear Support Vector Machine	0.87	0.500 0.800	0.89 0.88	0.85 0.90	0.53 0.41	0.65 0.57
Logistic Regression	0.87	0.425 0.750	0.89 0.88	0.81 0.90	0.58 0.44	0.68 0.59
Random Forest	0.86	0.600 0.700	0.89 0.88	0.86 0.92	0.48 0.39	0.62 0.55
Bernoulli Naïve Bayes	0.86	0.700 0.900	0.89 0.86	0.80 0.84	0.55 0.34	0.65 0.49
XGBoost	0.87	0.500 0.800	0.89 0.88	0.85 0.91	0.54 0.39	0.66 0.55
ogRAT	0.81	0.010 0.020	0.87 0.87	0.86 0.91	0.41 0.33	0.56 0.49

**Table 3 cancers-14-05023-t003:** Performance of machine-learning-based probabilistic classifiers on test dataset (results shown for some of the best-performing techniques: Support Vector Machine with linear kernel and Logistic Regression with stochastic average gradient solver) in comparison with oesophago-gastric cancer risk-assessment tool (ogRAT) matrix (performance calculated on same test set) for prediction of oesophago-gastric cancer incidence. Performances shown when making comparisons using the exact precision values achieved by the ogRAT.

Classifier	AUROC	Classification Threshold	Accuracy	Precision	Recall	F1
Linear Support Vector Machine	0.87	0.56 0.82	0.89 0.88	0.86 0.91	0.52 0.41	0.65 0.57
Logistic Regression	0.87	0.57 0.82	0.89 0.88	0.86 0.91	0.51 0.41	0.64 0.57
ogRAT	0.81	0.01 0.02	0.87 0.87	0.86 0.91	0.41 0.33	0.56 0.49

## Data Availability

The data are not publicly available due to restricted access.

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
