# Peer review of "Machine Learning for Risk Prediction of Oesophago-Gastric Cancer in Primary Care: Comparison with Existing Risk-Assessment Tools"

_cancers, 2022, doi:10.3390/cancers14205023_

Round 1

Reviewer 1 Report

The manuscript gives a good literature review of current limitation in oesophago-gastric cancer diagnose and the potential advantage to improve the diagnostic accuracy through machine learning. The study sample is well described. The data analysis methods, including how data were pre-processed, how model hyperparameters were turned, and what criteria were used in prediction comparison, were reasonable and clearly described. The conclusion was well supported by the data. The study results could have potential impact on future oesophago-gastric cancer diagnose. Overall, the manuscript is well written. 

I have one minor question. It is unclear how threshold values in Table S3 were determined. Were they suggested by the training set analysis results?

I suppose Table 2 reports results from test set data. Please make it clear in the table title.

Author Response

Response to Reviewer’s Comments

We would like to take this opportunity to thank the editors for giving us the chance to revise and resubmit this journal article. To the reviewer - thank you for taking the time to review the article and for the feedback provided. We feel that your suggestions have been helpful in improving the article and aiding its clarity and focus.  

Reviewer 1

The manuscript gives a good literature review of current limitation in oesophago-gastric cancer diagnose and the potential advantage to improve the diagnostic accuracy through machine learning. The study sample is well described. The data analysis methods, including how data were pre-processed, how model hyperparameters were turned, and what criteria were used in prediction comparison, were reasonable and clearly described. The conclusion was well supported by the data. The study results could have potential impact on future oesophago-gastric cancer diagnose. Overall, the manuscript is well written. 

Thank you for your comments highlighting the strengths of the study and we appreciate your feedback.

I have one minor question. It is unclear how threshold values in Table S3 were determined. Were they suggested by the training set analysis results?

Thank you for your point, this is a great question. Regarding the threshold values in Table S3, the range from 0.3 to 0.8 was chosen since this seems like a reasonable, wide range of threshold values at which considering a cut-off point for high risk of cancer might be feasible. The training set analysis helped determine this, yes, and model accuracies tended to tail off at lower or higher thresholds than this. Furthermore, lower thresholds yielded much lower precision values not comparable with the ogRAT and higher thresholds yielded much lower recall values which again, would not be comparable with the ogRAT tool. As stated in the paper, the study was designed with a view to bettering the performance of the ogRAT tool. It is unlikely – though not impossible – that threshold values outside of this range would be desirable. Thank you for taking the time to address this, and we hope this sufficiently answers your question.

I suppose Table 2 reports results from test set data. Please make it clear in the table title.

Thank you for your comment – this has now been amended in the revised version of the manuscript and we appreciate your suggestion.

Reviewer 2 Report

Report on “Machine Learning for Risk Prediction of Oesophago-gastric Cancer in Primary Care: Comparison with Existing Risk Assessment Tools” by Emma Briggs et al.

The study analyzes how machine learning models can improve the diagnosis of oesophago-gastric cancer performance using electronic health records. The authors propose the use of 5 popular algorithms in this field. This is an interesting study that assesses the advantage of using electronic patient records in cancer prediction.

The case selection seems adequate using matching techniques. The database is also divided in an appropriate way to train the models and later validate it. Continuous variables are dichotomized for a better comparison of their predictive power, although this implies a loss in their predictive capacity. The selection of initial parameters to train the models is well described in the supplementary material and the optimal parameters are also reported. In my opinion, this whole process is well detailed in the manuscript.

My main concern lies in the validation of the model and the lack of clinical utility analyzed of the best models. The interest of the study should be described or valued, is it to detect the largest number of cancers?, or a model that classifies as cancer those patients that really have a very high probability of having cancer. The best models have been selected for having a larger area under the ROC curve, but in practice the models with greater precision or recall may be those of interest.

In my opinion table 2 is not well designed. If I am thinking in a screening program, perhaps I may be interested in models with a high recall value, 95%, 90%, 85% or 85%, in this case we must choose as best model the one that has the greater precision for a fix recall value. It is true that the purpose of the study is to compare with existing tools, well, in this case it will be necessary to take as reference the fixed value of recall or precision of the ogRAT model and present the value of the MLR models for that same value. In this way it will be possible to assess the improvement of the models. Therefore, the comparison of the models must be done for the same reference value of precision or recall.

Also, in my opinion there are not so many results in the study, I would like to know the results of all the models, for example, the random forest model has an AUC very similar to the logistic regression. In table 2 can be incorporated values for all MLR models.

I would also appreciate further developments in the discussion section for the clinical interest of the application of these tools and the possible limitations on data collection for the inputs of the model.

Author Response

Response to Reviewer’s Comments

We would like to take this opportunity to thank the editors for giving us the chance to revise and resubmit this journal article. To the reviewer - thank you for taking the time to review the article and for the feedback provided. We feel that your suggestions have been helpful in improving the article and aiding its clarity and focus. 

Reviewer 2

The study analyzes how machine learning models can improve the diagnosis of oesophago-gastric cancer performance using electronic health records. The authors propose the use of 5 popular algorithms in this field. This is an interesting study that assesses the advantage of using electronic patient records in cancer prediction.

The case selection seems adequate using matching techniques. The database is also divided in an appropriate way to train the models and later validate it. Continuous variables are dichotomized for a better comparison of their predictive power, although this implies a loss in their predictive capacity. The selection of initial parameters to train the models is well described in the supplementary material and the optimal parameters are also reported. In my opinion, this whole process is well detailed in the manuscript.

Thank you for your comments – we appreciate your opinion and the time you’ve taken to assess the main strengths and limitations of the study.

My main concern lies in the validation of the model and the lack of clinical utility analyzed of the best models. The interest of the study should be described or valued, is it to detect the largest number of cancers?, or a model that classifies as cancer those patients that really have a very high probability of having cancer. The best models have been selected for having a larger area under the ROC curve, but in practice the models with greater precision or recall may be those of interest.

Thank you for your question – we agree with what you’ve said. The interest of the study is to improve upon the performance of a current cancer risk assessment tool (the ogRAT) in terms of overall accuracy and the emphasis is on being able to detect a greater number of cancer cases (i.e., improving upon the current recall rate). The study seeks to address the issue of being able to more strategically refer patients for further investigation i.e., detecting more cases with little to no cost to the false positive rate, in order to concentrate resources where they are most needed and keep costs to a minimum. We attempted to highlight this in our introduction; however, it is clear that further discussion of this is required. In light of your comments, additional explanation behind the rationale of the study in terms of model performance has been given in the Introduction (lines 113-118) and is as follows:

“The study seeks to address the question as to whether ML models could help with the strategic referral of patients for further investigation and the earlier detection of a greater number of cancer cases. In terms of model performance, this translates to the extent to which recall values might be increased whilst roughly preserving precision values, thereby detecting a larger number of cases yet minimising the number of unnecessary procedures.”

This is also emphasized in the results section, which discusses model performances in terms of these aims.

Thank you for helping us to revise this section in order to improve the clarity of the study’s aims, and we hope that this sufficiently addresses your question.

In my opinion table 2 is not well designed. If I am thinking in a screening program, perhaps I may be interested in models with a high recall value, 95%, 90%, 85% or 85%, in this case we must choose as best model the one that has the greater precision for a fix recall value. It is true that the purpose of the study is to compare with existing tools, well, in this case it will be necessary to take as reference the fixed value of recall or precision of the ogRAT model and present the value of the MLR models for that same value. In this way it will be possible to assess the improvement of the models. Therefore, the comparison of the models must be done for the same reference value of precision or recall.

Also, in my opinion there are not so many results in the study, I would like to know the results of all the models, for example, the random forest model has an AUC very similar to the logistic regression. In table 2 can be incorporated values for all MLR models.

Thank you again for your comments. The original values in Table 2 were given to represent model performances which were comparable to the ogRAT in terms of precision so that associated increases in recall could be commented on; however, following your comments we see that this was not originally explicitly clear and have altered the manuscript accordingly.

Taking your suggestion into consideration, we have therefore amended Table 2 to include performances for all models, which makes sense given the similarity in model accuracies. We have also added an additional table – Table 3 – which takes the fixed reference values for precision achieved by the ogRAT and displays an example using two of the models (Linear Support Vector Machine and Logistic Regression) to show that ML can exactly conserve the precision rate of the ogRAT and simultaneously offer a clear improvement in terms of recall. This also aligns with the main rationale behind the study of detecting a greater number of cancer cases earlier and therefore we feel that this was a valuable improvement in the revision – thank you again for your help. Again, we hope that this sufficiently addresses your comments. For further results of all models across a wider range of thresholds and stratified by demographics (age, sex, cancer site), please see the supplementary material.

I would also appreciate further developments in the discussion section for the clinical interest of the application of these tools and the possible limitations on data collection for the inputs of the model.

Thank you – we think that this is another great point and have therefore included an additional paragraph in the discussion section which we hope reflects your suggestion. This is as follows (lines 317-324):

“Such a tool would, for example, take the form of a digitised risk prediction tool embedded in the primary care clinician’s workflow. Based on symptomatic and lab test information input with clinical codes such as SNOMED or Read codes, the ML algorithm would calculate a cancer risk score for each patient and subsequently flag patients with a high-risk score according to a predetermined risk threshold. Implementing the tools in this way does risk bias related to coding inconsistencies and unreported symptoms, for example, and future work should seek to address the mitigation of such biases.”